# GENERATING PRAGMATIC EXAMPLES TO TRAIN NEURAL PROGRAM SYNTHESIZERS

**Saujas Vaduguru**
Carnegie Mellon University
svadugur@cs.cmu.edu

**Daniel Fried**
Carnegie Mellon University
dfried@cs.cmu.edu

**Yewen Pu**
Autodesk Research
yewen.pu@autodesk.com

## ABSTRACT

Programming-by-example is the task of synthesizing a program that is consistent with a set of user-provided input-output examples. As examples are often an under-specification of one's intent, a good synthesizer must choose the intended program from the many that are consistent with the given set of examples. Prior work frames program synthesis as a cooperative game between a listener (that synthesizes programs) and a speaker (a user choosing examples), and shows that models of computational pragmatic inference are effective in choosing the user intended programs. However, these models require counterfactual reasoning over a large set of programs and examples, which is infeasible in realistic program spaces. In this paper, we propose PRAX, a novel way to amortize this search with neural networks. We sample pairs of programs and examples via self-play between listener and speaker models, and use pragmatic inference to choose informative training examples from this sample. We then use the informative dataset to train models to improve the synthesizer's ability to disambiguate user-provided examples *without human supervision*. We validate PRAX on the challenging task of synthesizing regular expressions from example strings, and find that our method (1) outperforms models trained without choosing pragmatic examples by 23% (a 51% relative increase) (2) matches the performance of supervised learning on a dataset of pragmatic examples provided by humans, despite using no human data in training.

## 1 INTRODUCTION

In program synthesis – specifically programming-by example-(PBE) – a user describes a target program using input-output examples (i.e. test cases) and the synthesizer finds a program that is consistent with these input-output examples. In PBE, the users directly express the semantics of the intended program (what it *should do*) without having to understand the syntax of the program (what it *should look like*). Such systems have found real-world use in a variety of scenarios such as spreadsheet formulas (Chen et al., 2021; Gulwani, 2011) and data wrangling (Feng et al., 2018).

An important aspect of inferring programs from examples is dealing with *ambiguity*. Given a set of examples, there can be many spurious programs consistent with the set, and picking out the right one the user has in mind is a long-standing challenge. For example, when describing the regular expression a+b*,[1] an informative user might provide the example $(ab, \checkmark)$ indicating that the string *ab* matches the target regular expression. However, to the program synthesizer, both a+b* and a*b+c?[2] would be among many acceptable answers based on this example.

Pu et al. (2020) resolves this ambiguity by framing program synthesis as a coorporative communicative game: the user chooses an informative set of examples to convey the program to the synthesizer, and the synthesizer chooses a program under the assumption that these examples were chosen informatively. Models of *pragmatic inference*, specifically, the Rational Speech Acts (RSA) framework (Frank & Goodman, 2012) can then be used to build a program synthesizer that can resolve ambiguity via recursive Bayesian inference. The RSA framework allows the synthesizer to reason about

---

[1] 1 or more *a* s followed by 0 or more *b* s

[2] c? means optionally having a *c* at the end

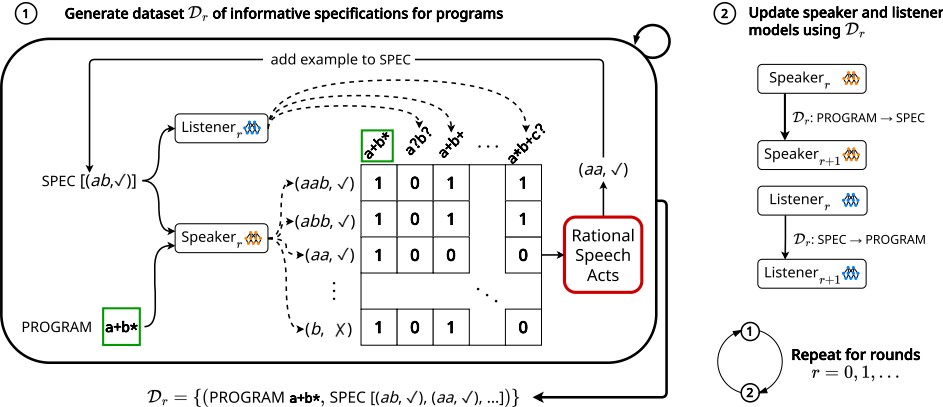

Figure 1: PRAX iteratively generates datasets containing increasingly informative program specifications (lists of examples consistent with the program), and updates models on the generated datasets. ① We use a Speaker model — that generates an example consistent with a target PROGRAM — to propose a set of candidate specifications. Using the Rational Speech Acts model of pragmatic reasoning (red box; described in in Figure 2), we choose the example that is most informative to a Listener model that synthesizes programs consistent with a given specification. In this manner, we incrementally build the list of examples SPEC for the PROGRAM. We repeat this for different programs to create a dataset of informative PROGRAM-SPEC pairs. ② We use the dataset to update the Speaker and Listener models. We train the speaker to generate the selected pragmatic examples, and the listener to synthesize the target program given the generated examples.

what program a user intended, given that they chose that particular set of examples rather than a different one. For example, the synthesizer could reason that a user that wanted to describe a*b+c? would have chosen an example containing the character c. However, this approach requires the synthesizer to perform expensive counterfactual inference over the space of all programs and examples to resolve ambiguity, making it difficult to scale to realistic programming domains.

To scale to realistic program spaces, modern approaches of PBE systems have relied on training neural networks to efficiently search through large program spaces for consistent programs given examples (Balog et al., 2017; Devlin et al., 2017). In this paper, we explore whether we can use simulated reasoning in communication games using the RSA framework as a way to generate training data consisting of pragmatic examples. The generated data is then used to train neural networks to enable scaleable pragmatic program synthesis. We dub this approach PRAX. We hypothesize that since the RSA framework computationally models how a human chooses examples to communicate a program, end users would succeed more often when communicating with a neural synthesizer trained on pragmatic data (our work) as compared to a neural synthesizer trained on non-pragmatic data (Balog et al., 2017; Devlin et al., 2017).

An overview of PRAX is shown in Figure 1. We start with a neural *literal listener* — a synthesizer trained in the style of Devlin et al. (2017) — and a neural *literal speaker* that generates examples consistent with a given program. We generate a sequence of pragmatic examples incrementally to obtain a training pair (program, examples). This pair is then added to an aggregate training set, which is used to finetune both the speaker model — making it more likely to generate pragmatic examples — and the listener model — making it more likely to recover the intended program given pragmatic examples.

We validate the effectiveness of PRAX on the well-studied PBE task of inferring regular expressions from a set of examples. Each example is a pair (*string*, **bool**), indicating whether a particular string matches the regex. To compare our training algorithm to standard supervised learning from human-annotated data, we collect a novel dataset of human annotations, consisting of (program, examples) where the examples were given by a person for a total of 440 regular expressions. We find that with only a small number (40) of human annotations – used only for model selection – our method is able to outperform a system that is fine-tuned using 400 annotated regexes from this dataset. We conduct human evaluation of PRAX by giving a user a target regex to communicate interactively to

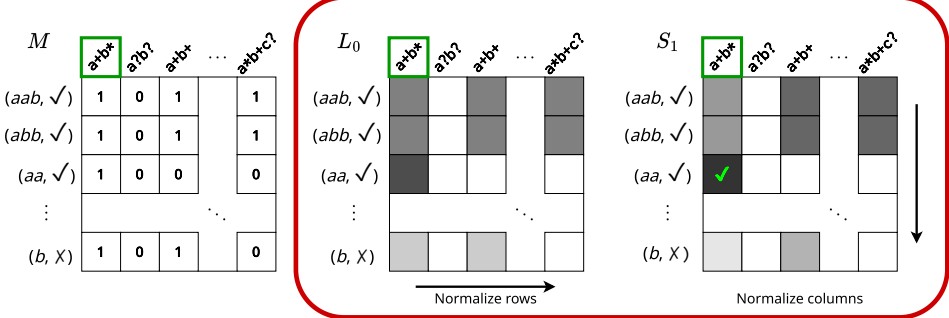

Figure 2: An illustration of how the Rational Speech Acts framework is used to select an informative example for a given program. We start with the matrix corresponding to the consistency relation between the sample of programs and examples shown in Figure 1. We obtain a literal listener distribution $L_0$ over programs for each example by normalizing the rows of this matrix. Since the $M$ matrix is binary, each row in $L_0$ is a uniform distribution over consistent programs in the sample — any of the consistent programs is equally likely to be the intended program. We then obtain a pragmatic speaker distribution $S_1$ by normalizing the columns of the $L_0$ matrix: modeling the probability an informative speaker might have for choosing each example when communicating a program to a literal listener. RSA outputs the highest-probability example in $S_1$ (e.g., (*aa*, ✓)) in the column corresponding to the target program (e.g., a+b*).

the synthesizer using examples, and find that the informative examples generated by our procedure substantially improve the performance of a regular expression synthesizer, with improvements of 22.8% absolute (51.4% relative) in accuracy (11 participants, 340 regexes total). PRAX, despite not using human-provided data during training, matches the performance of of a model fine-tuned on a large dataset of human-written pragmatic examples. Our code and data are available at `https://github.com/saujasv/generating-pragmatic-examples`.

## 2 BACKGROUND

**Programming-by-Example** In this paper, we tackle the task of finding a program $\in P$, where $P$ is a space of possible programs. As a *specification* of intent, a user provides a sequence of input-output examples $\in E^+$,[3] where $E = \mathcal{X} \times \mathcal{Y}$ is the space of all possible input-output pairs that programs in $P$ operate over. For example, $P$ may be a space of regular expression programs, $\mathcal{X}$ the space of all strings in the alphabet that the regular expressions are defined over, and $\mathcal{Y} \in \{✓, ✗\}$ where output ✓ indicates whether the input string matches the regular expression. For simplicity of explanation, in this section we consider cases where the specification consists of *a single example*, deferring the cases of multiple examples to the next section.

The semantics of programs are captured by the *consistency relation* $M$ between $P$ and $E$:

$$M = \{(\text{program}, \text{example}) \mid \text{example} = (x, y) \in \mathcal{X} \times \mathcal{Y}, \text{program}(x) = y\}$$

A program is consistent with an example iff executing the program on the input produces the intended output. We can view $M$ as a *consistency matrix* where each row corresponds to an example, each column corresponds to a program, and an element is 1 if the program is consistent with the example and 0 otherwise (Figures 1 and 2, left).

**Literal model of program synthesis** A minimal requirement of a program synthesizer is that it finds *any* program that is consistent with the given specification. We refer to such a synthesizer as the *literal listener* $L_0$, which naively assigns equal probability to any consistent program.

$$L_0(\text{program}|\text{example}) \propto M(\text{example}, \text{program})P(\text{program}) \tag{1}$$

This literal listener distribution is given by normalizing the rows of the consistency matrix to produce uniform probability distributions ($L_0$ in Figure 2). However, this literal listener cannot resolve

---

[3]Here the notation $X^+$ indicates a sequence of 1 or more elements belonging to $X$

ambiguity when interacting with users, as it places equal probability on all consistent programs. A literal speaker – one that generates *any* consistent examples for a given program – is defined analogously as $S_0(\textsf{example}|\textsf{program}) \propto M(\textsf{example}, \textsf{program})P(\textsf{example})$.

**Pragmatic model of program synthesis**    When interacting with a synthesizer, users choose examples that are *informative* – those that distinguish the program they desire from others. For example, given the specification $[(\textit{ab}, \checkmark)]$, a user is likely wants the regular expression a+b+ than a+b+c?, since they probably would have included the character $c$ if they wanted the latter expression.

To leverage the informativity of examples, Pu et al. (2020) use the Rational Speech Acts (RSA; Frank & Goodman 2012) framework to derive a *pragmatic program synthesizer* that resolves ambiguity by modeling how people choose examples informatively. First, they construct a pragmatic speaker $S_1$ that chooses an example in proportion to the likelihood that it would make the literal listener infer the intended program:

$$S_1(\textsf{example}|\textsf{program}) \propto L_0(\textsf{program}|\textsf{example})P(\textsf{example}) \qquad (2)$$

Using a uniform prior $P(\textsf{example})$, the $S_1$ distribution is given by normalizing the columns of the $L_0$ matrix ($S_1$ in Figure 2). As we can see, given a program $S_1$ selects an example in proportion to the likelihood of $L_0$ recovering the program given the example, choosing examples that are informative to the listener.[4]

Finally, a pragmatic listener (program synthesizer) $L_1$ is built on top of $S_1$:

$$L_1(\textsf{program}|\textsf{example}) \propto S_1(\textsf{example}|\textsf{program})P(\textsf{program}) \qquad (3)$$

using a prior $P(\textsf{program})$ over programs. This listener resolves ambiguity by choosing that program that an informative speaker (modeled by $S_1$) would have described using the chosen example.

Pu et al. (2020) demonstrated that building a pragmatic synthesizer $L_1$ in this way allows for users to communicate the target program to the synthesizer using fewer examples without training a model on human-produced examples or explicitly defining a prior over the space of programs. However, in realistic domains, enumerating large numbers of programs and examples is intractable, preventing the application of this framework to a broader range of tasks.

## 3 METHOD

In this section, we describe the iterative process by which we bootstrap a pragmatic neural program synthesizer by generating informative specifications, without human supervision (Figure 1). The full algorithm is detailed in Appendix C.

### 3.1 SPEAKER AND LISTENER MODELS

We build on past work that uses neural models as specification-conditioned proposal distributions over programs (Balog et al., 2017; Devlin et al., 2017). Our listener (synthesizer) models represent distributions over programs $L_\theta(\textsf{program}|\textsf{examples})$. Our speaker (specification generation) models generate the sequence of examples in a specification autoregressively: $S_\phi(\textsf{example}_i|\textsf{program}, \textsf{examples}_{1:i-1})$.[5] While all our listener and speaker models share the same architecture and initialization, we vary their training data, as described below.

### 3.2 TRAINING BASE MODELS

As a foundation for our approach, we train *base* listener and speaker models to approximate literal speakers and listeners $S_0$ and $L_0$ (Equation (1)). Since we cannot enumerate the consistency matrix completely and normalize rows, we obtain these approximate models by training on data obtained by randomly sampling an input from the space of inputs $\mathcal{X}$, and executing the program on the input

---

[4]For a sequence of examples, Pu et al. (2020) propose factoring the pragmatic speaker distribution autoregressively as $S_1(\textsf{examples}|\textsf{program}) = \prod_{i=1}^{N_{examples}} S_1(\textsf{example}_i|\textsf{program}, \textsf{examples}_{:i})$

[5]We train the speaker to predict the input-output pair to encourage the model to capture aspects of the domain semantics

to obtain the output (e.g., by checking if an sampled example string is matched by a sampled regular expression). This lets us generate as many samples from $M$ as we can, which we can use to train a base listener to approximate $L_0$ (Equation (1)), and a base listener to approximate an analogous $S_0$, using standard maximum likelihood training. This is essentially the method proposed by Devlin et al. (2017). We denote the resulting base listener model as $L_{\theta_0}$ and the base speaker model as $S_{\phi_0}$, and use these as the initial models in our iterative model bootstrapping procedure.

### 3.3 GENERATING INFORMATIVE EXAMPLES

The crux of our algorithm is using the existing $S_\phi$ and $L_\theta$ to approximate $S_1$, which can then be used to generate training data to improve $S_\phi$ and $L_\theta$ over rounds of training. At each round $r$ of our approach, we use the current speaker and listener models, together with the RSA procedure, to create a dataset of informative examples specifying programs (① of Figure 1). We incrementally generate examples to create a specification. Given a partial specification of $i$ examples $\mathsf{examples}_{1:i}$, we sample a set of additional candidate examples: $S_{\phi_r}(\mathsf{example}_{i+1}|\mathsf{program}, \mathsf{examples}_{1:i})$. Similarly, we sample a set of alternative programs: $L_{\theta_r}(\mathsf{program}|\mathsf{examples}_{1:i})$ from the partial specification.[6]

We can then compute the sampled consistency matrix over the generated examples and programs, and use RSA inference as shown in Figure 2 to choose the highest scoring example from the approximate $S_1$ distribution.[7] This example is added to the partial specification, and we repeat until a maximum number of examples are reached.[8] The completed program-specification pair is then added to a dataset $\mathcal{D}_r$ of examples from that round of training. This process amounts to choosing an example proposed by the current speaker model that minimizes ambiguity among programs that the current listener infers to be likely. The full algorithm for incrementally generating a sequence of examples is presented in Algorithm 2 (Appendix C).

### 3.4 MODEL UPDATES

We use the dataset $\mathcal{D}_r$ to update both the speaker and listener models as sketched in part ② of Figure 1 using standard maximum likelihood training. In each round $r$ we further fine-tune the speaker and listener models on the generated data to obtain the updated parameters $\theta_{r+1}$ and $\phi_{r+1}$. The full algorithm to iteratively generate examples and update the model is presented in Algorithm 1 (Appendix C). To select the model that works best with human-provided examples, we choose the model that maximizes a model selection metric computed over a small set of programs paired with human-provided examples.[9] Note that this validation set is never used to update the model parameters, and only is used to choose a model.

## 4 EXPERIMENTS

### 4.1 REGULAR EXPRESSIONS

We validate the training algorithm we propose on the task of synthesizing regular expressions ('regexes') as formally defined in Section 2. We use the regular expression domain-specific language presented by Ye et al. (2020). In addition to defining a regular expression specification language, they also define a sampling distribution over the space of regular expressions that we use to sample programs for training and evaluating our model. This distribution uses templates that generalize types of regular expressions that people ask about on fora such as StackOverflow. Further details are provided in Appendix A.

---

[6]We follow prior work (Pu et al., 2020) and impose a *uniform*, rather than learned, prior over this sample.

[7]Ideally, one could perform exact inference to draw samples from $S_1$ directly. As stated earlier, this is intractable. Therefore, we first sample a subset of the rows and columns in the consistency matrix, then perform the RSA inference over this much smaller and denser sampled matrix. However, the consistency matrix $M$ is sparse (mostly 0s) – most programs are inconsistent with any non-trivial set of examples – allowing for reasoning about a sample of the matrix.

[8]Since the models are used only to generate the programs and utterances that are used to create the lexicon, we can draw examples from other sources, including models other than $S_{\phi_r}(\mathsf{example}_i|\mathsf{program}, \mathsf{examples}_{:i})$ and $L_{\theta_r}(\mathsf{program}|\mathsf{examples}_{:i})$.

[9]This amounts to performing early stopping on the validation metric.

## 4.2 Models for comparison

**Base models**   We use ByT5-small models (Xue et al., 2022) as the backbone for all speaker and listener models. To obtain the base speaker $S_{\theta_0}$ and listener $L_{\phi_0}$ models that approximate the literal speaker and listener respectively, we use a set of 300,000 randomly-generated program–specification pairs (with varying numbers of examples in each specification) and finetune the pretrained ByT5 checkpoint. Full details of training are provided in Appendix B. The $L_{\theta_0}$ model acts as the LITERAL model in our experiments.

**PRAX**   We start with the base models $S_{\theta_0}$ and $L_{\phi_0}$ and use the iterative data generation and fine-tuning algorithm to obtain a sequence of synthesis models $L_{\phi_r}$ for rounds $r = 0, \ldots, R_{max}$. We use the TOP-1 metric evaluated on a small validation set to choose the best model which we refer to as the PRAX model.

**Finetuning on human-provided specifications**   We obtain an HFT model by fine-tuning $L_{\phi_0}$ on a curated high-quality human-provided specifications (Section 4.5). This model allows us to compare how well our approach of using model generated informative examples compares with sourcing more expensive human annotations.

**GPT-3.5**   We evaluate GPT-3.5 by using the program-specification pairs we obtain as users interact with the other three models, and revealing each specification to the GPT-3.5 model in the order the user provided them, one example at a time. We stop when the model guesses the correct regular expression, or when all the examples are presented. We can think of this as a form of interaction where the human doesn't observe the outputs of this model while giving examples. Further details of how the model is prompted are in Appendix F.

**Inference**   Crucial to our approach is the ability to generate programs from examples, and vice versa. To generate programs from a specification (sequence of examples from either self-play or human), we present it to the listener, and sample 500 programs using top-$p$ sampling (Holtzman et al., 2020) with $p = 0.9$. We then deduplicate the set of sampled programs, and filter out programs inconsistent with the given specification. We can then sort the remaining programs by their score under the model to obtain a ranked list of consistent programs. Similarly, to generate specifications from a program, we sample 500 examples using top-$p$ sampling with $p = 1$ and check that the examples are consistent with the program.

## 4.3 Procedure

We evaluate the model on the basis of successful communications on 11 human participants. A sampled regex $p$ is given to a human participant, whom describes it using a sequence of examples, providing one example in each turn for upto a maximum of 10 turns. The synthesizer takes the examples provided and generates a ranked list of inferred programs, of which the top-1 regex $p'$ is shown to the participant. The communication is successful when $p = p'$, at which point the interaction ends.[10] A total three synthesizers were considered – the LITERAL model, the human fine-tuned HFT model, and the PRAX model. The identity of the models is referred to the users only as differently colored robots. The study yielded communication history over a total of 340 regexes (109 to the LITERAL model, 113 to the HFT model, and 118 to the PRAX model).[11] Further details about how the user study is conducted are provided in Appendix E.

## 4.4 Measurement

We consider the following metrics. TOP-1@$t$ measures whether the model's top-1 matches intended regular expression at any point at turn $t$ of the interaction. We can also consider the average value of TOP-1 by aggregating across the turns $t \in \{1, \ldots, 10\}$. Averaging across turns rewards models that pass success criteria given fewer turns — a model that can infer the target regex at turn 4 on average

---

[10]We used the `greenery` Python library to identify regex matches in terms of semantic similarity, and not just surface form match.

[11]a small bug caused us to collect few extra interactions for some of the models, which does not change the measurements on the models' relative performances

| Model | TOP-1@10 (SE) | TOP-10@10 (SE) | EDIT DISTANCE $\leq 1$@10 (SE) |
|---|---|---|---|
| LITERAL | 0.434 (0.047) | 0.522 (0.047) | 0.513 (0.047) |
| GPT-3.5[*] | 0.074 (0.014) | 0.189 (0.021) | 0.082 (0.015) |
| HFT | 0.587 (0.047) | 0.623 (0.047) | 0.614 (0.047) |
| PRAX | **0.661** (0.043) | **0.703** (0.042) | **0.694** (0.042) |

Table 1: Success metrics at the end of 10 turns of interaction with each model, with standard errors computed using bootstrap sampling. [*] indicates that the results are in replay.

| Model | TOP-1 (SE) | TOP-10 (SE) | EDIT DISTANCE $\leq 1$ (SE) |
|---|---|---|---|
| LITERAL | 0.233 (0.028) | 0.333 (0.034) | 0.296 (0.031) |
| GPT-3.5[*] | 0.048 (0.010) | 0.122 (0.014) | 0.056 (0.011) |
| HFT | 0.349 (0.032) | 0.424 (0.035) | 0.390 (0.034) |
| PRAX | **0.373** (0.028) | **0.430** (0.030) | **0.432** (0.031) |

Table 2: Average success metric over 10 turns. We see that the PRAX training method we propose is on par with HFT, and outperforms other baselines significantly for all success criteria. [*] indicates that the results are in replay.

is better than a model that can only infer the target at turn 10. We use TOP-1 over a validation set as the model selection criterion for our proposed method. TOP-10@$t$ and TOP-10 are similarly defined. EDIT DISTANCE $\leq 1$@$t$ measures whether the model's highest scoring prediction in any of the turns up to $t$ is at most a 1 token edit from the intended program, and EDIT DISTANCE $\leq 1$ is the average of this value over $t \in \{1, \ldots, 10\}$.

## 4.5 HUMAN-PROVIDED SPECIFICATIONS

An alternative to generating informative examples using the method we propose is to have human annotators provide examples. We collect a new dataset of high-quality program-specification pairs.

**Procedure** We present a participant with a sampled regular expression, and instruct them to provide examples that they might use to illustrate the regular expression to another person. Participants are asked to provide at least 5-7 examples. We verify whether the examples are informative by checking whether a different annotator is able to identify the program which the given set of examples describes. Further details about the data collection process are presented in Appendix D.

**Usage of data** We collect a total of 440 program-specification pairs. We sample a small subset of 40 pairs that received 2 "correct" verifications as a validation set for model selection. We use the other 400 pairs as a training set to finetune the $L_{\theta_0}$ models on human-provided informative examples, obtaining HFT (see Appendix B).

## 4.6 RESULTS

Table 1 shows the rate of success for different models at the end of 10 turns of interaction. We see that training on informatively examples results in large gains in performance. Both the PRAX and HFT models significantly outperform the literal model for all three criteria of success. Looking at the aggregate success rate across turns in Table 2 reveals that it is not just that the PRAX synthesizer eventually catches up to the HFT model, but also performs on par with it over the course of the interaction. Figure 3 shows the progression of each metric over the course of interaction. In contrast, we see that GPT-3.5 performs worse than the literal model. One reason for this could be that the distribution of regular expressions that the GPT-3.5 encounters in its training data could be quite different, leading to worse performance. In conclusion, the experiments validate our hypothesis that humans communicate more effectively with the model trained on informative examples (the PRAX and HFT models) than with a model trained on randomly chosen examples (the LITERAL model).

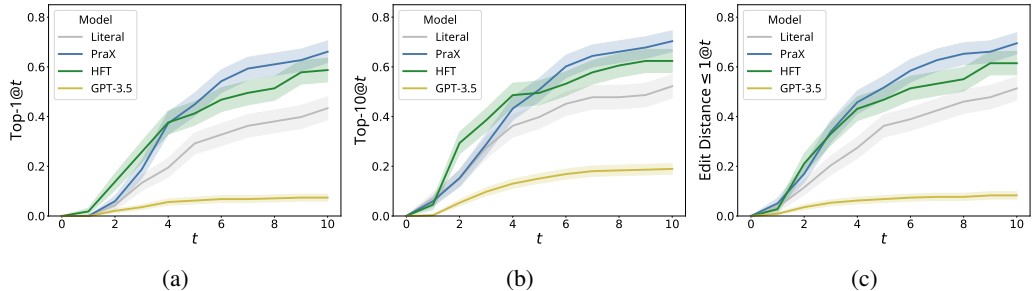

(a)          (b)          (c)

Figure 3: Performance of various models as a function of turns, measured in (a) TOP-1@$t$, (b) TOP-1@$t$, and (c) EDIT DISTANCE $\leq 1$@$t$. Lines show averages, and bands are standard errors. Our model PRAX, trained entirely from self-play and RSA inference without using human-provided data performs better than the non-pragmatic LITERAL model across all turns and metrics, and matches the performance of HFT tuned on a human-provided examples.

| Target program | Examples | LITERAL | PRAX |
|---|---|---|---|
| 4A{2,} | 4AAAAAAAAAAAAAAAAAA ✓
4AA ✓ | 4A{1,} | 4A{2,} |
| [A-Z]{1,}i{2,4} | Aiii ✓
Bii ✓
BBiiii ✓
AAAAAAAAAii ✓ | (A{1,}\|B{1,})i{2,4} | [A-Z]{1,}i{2,4} |

Figure 4: Example specifications for two programs provided during the user study, along with the highest ranked guess from the LITERAL and the PRAX models.

**Examples of synthesized programs**   Figure 4 shows examples of guesses by the LITERAL and PRAX models given the same sequence of examples. In the first case, we see that the PRAX model is able to infer that if a user wanted a regular expression that accepted *4A*, they would have specified that, and instead correctly guesses that the user wanted at least two *A* s in the string. The second example also shows how the LITERAL model synthesizes a regular expression that is correct, but is too specific, while the PRAX model recovers the correct generalization.

## 5   ANALYSIS OF TRAINING

Figure 5 shows the progression of the TOP-1 metric over the course of different rounds of training. We see that as we train the model for more rounds, the performance of the model generally increases, and then tapers off. This shows that as the model is trained for more rounds, it gets increasingly pragmatic. Since the model we choose for the user study is trained for 5 rounds, on $5 \times 1024 = 5120$ programs, we also compare to training the model for only a single round on the same number of programs. In a replay study (similar to how we evaluated GPT-3.5; Figure 5), we find that iteratively generating data and updating the model performs better. We also see that finetuning the base model on 400 examples (to match the HFT setting) from a later round of training also results in a strong model, suggesting that as the speaker is trained more, it generates examples that are useful to finetune a listener model.

## 6   RELATED WORK

**Neural network models of pragmatic reasoning**   Prior work has applied the the RSA pragmatic reasoning framework to improve neural models at inference time for tasks including image caption-ing (Andreas & Klein, 2016; Cohn-Gordon et al., 2018), instruction generation (Fried et al., 2018a), vision-and-language navigation (Fried et al., 2018b), and machine translation (Cohn-Gordon & Goodman, 2019). RSA is used at inference time to re-rank multiple outputs from the neural models.

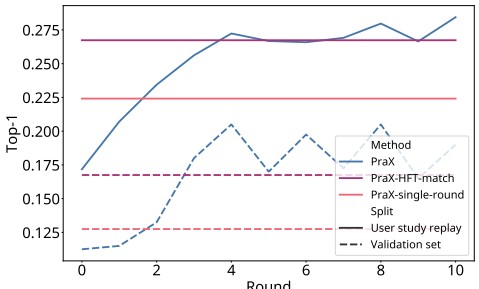

Figure 5: TOP-1 metric over the course of rounds of training of the PRAX model. We report the metric on the validaton set as well evaluating on all interactions from the user study in the replay setting (similar to how we evaluated GPT-3.5). We compare the accuracy over rounds of training to generating specifications and updating the models only once, amounting to a single round of the procedure with more programs (PRAX-single-round). We also compare to fine-tuning the base model on 400 pairs (same number as HFT) generated by the speaker in the 5th round of training (PRAX-HFT-match) to assess the quality of our speaker-generated examples.

PRAX has two advantages over these works: (1) it requires no human-provided data during training; (2) it uses RSA at training time via data generation, amortizing expensive RSA computaion.

Other approaches have used pragmatically-motivated training procedures. The closest works to ours are White et al. (2020) and Lazaridou et al. (2020), who use reinforcement learning approaches to fine-tune a speaker model using reward from a fixed listener model. Monroe & Potts (2015) and McDowell & Goodman (2019) backpropagate through the RSA procedure at training time to reason counterfactually about pragmatically produced utterances from humans. Again, PRAX is unique in that it does not require human-provided training data.

Finally, Andreas & Klein (2016) find that amortizing pragmatic reasoning during training does *not* perform as well as explicit pragmatic reasoning during inference in the domain of image captioning. We demonstrate that amortization is in fact effective for the domain programming-by-example.

**Pragmatic reasoning for program synthesis**   Similar to our work, Vaithilingam et al. (2023) conduct a study of how users interact with an exact RSA pragmatic regular expression synthesizer over a toy domain of ∼1000 regexes total over strings of only 0s and 1s. Pu et al. (2023) propose a way to make pragmatic PBE more efficient by inferring a global ranking function, but still relies on the expensive exact RSA during training. Our approach is different in that by using neural models for speakers and listeners *at training time*, we are able to scale to a realistic regex domain. Pertseva et al. also present an approach to version space algebra-based approach to regular expression synthesis from examples by explicitly modeling the probability of examples describing programs (as in our speaker models), but they work with only positive examples (a subset of our example space with examples that have the output ✓, excluding those with the output ✗). Ferreira et al. (2021) present an SMT-based method that reasons about distinguishing inputs to synthesize regular expressions. We discuss connections to iterated bootstrapped training for program synthesis in Appendix H.

## 7   CONCLUSION

We present PRAX, a novel algorithm that bootstraps pragmatic program synthesizers by (1) generating datasets using self-play between a speaker (program → examples) and a listener model (examples → programs), and (2) training on the generated data. Crucial to our approach is the use of *pragmatic inference* to make the generated data more informative. PRAX produces pragmatic program synthesizers with minimal supervision: in a challenging regular expression domain, matching the performance of synthesizers fine-tuned on human-produced examples, despite not using any human-provided data during training. Future work might explore scaling pragmatic program synthesis to open-ended Python code generation, and application to multimodal specifications — e.g. with natural language and examples (Ye et al., 2020).

ETHICS STATEMENT

Our dataset collection process and user study constitute human subjects research. Our studies were deemed exempt from full IRB review by our institution. All participation was voluntary. Participants signed an online consent form, and were compensated fairly for their time.

ACKNOWLEDGEMENTS

The authors would like to thank Xi Ye for help with sampling regular expression programs, Eric Lu and Kevin Ellis for initial discussions, Priyan Vaithilingam for inputs on the interface and user study, Kira Jones for help with compensating participants, Catherine Copetas for help with advertising, Alex Xie and Simran Khanuja for help with testing the user study interface, and Vijay Viswanathan, Jared Fernandez, Zhiruo Wang, Harshita Diddee, and Lindia Tjuatja for feedback on drafts. SV was supported by a gift from Autodesk Research.

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

## A  PROGRAM SAMPLING

We sample programs from the CONCATENATION and SEPARATION templates from Ye et al. (2020) since the INTERSECTION template is not supported by some popular regular expression libraries like Python's standard re module. We sample programs with a maximum concatentation depth of 3, and sample programs from the CONCATENATION and SEPARATION templates in a 5:1 ratio using the sampling tools provided by Ye et al. (2020).

Despite controlling the complexity of programs during sampling, many sampled regular expressions are quite long and difficult for a human to reason about easily. To make the task easier for annotators and user study participants, we use the number of tokens in the regular expression program as a proxy for complexity, and cut off programs that are shown to humans at a maximum length of 30 tokens.

## B  TRAINING BASE MODELS

We use ByT5 pretrained models (Xue et al., 2022) as the backbone to build all our speaker and listener models. We sample programs as described in Appendix A and generate uninformatively chosen examples for the programs. We then train the models on a sequence-to-sequence task. The input to the listener is a linearized sequence of examples, and the output is the program. The input to the speaker is a program followed by a linearized sequence of prior examples, and the output is the next example in the sequence.

We generate random examples by first choosing whether the example is a positive or negative examples with equal probability. If the example is positive, we use the rstr Python package to sample strings that match the target regular expression. If the example is negative, we use the same package to sample a string from the regular expression .* and check that it doesn't match the target program.

We train the base models on 100,000 programs. For each program, we randomly sample 3 specifications. The length of each specification is chosen to be an integer between 0 and 15, uniformly at random. The models are trained for 1 epoch, using the AdamW optimizer with a learning rate of $5 \times 10^{-5}$, with the learning rate warmed up over the first 10% of training steps, and then decayed linearly. The batch size is set to 32.

Additionally, we found that training the base model (that we dub L0) for 3 epochs achieved higher performance. To ensure a fair comparison, we trained from the higher performing base model (that we dub L0+) using our PRAX algorithm. We see in Figure 7 that we get significant improvements over the L0+-based literal model even when continuing from an improved base model.

We also compare to training the base model from a randomly-initalized checkpoint Figure 6, and find that training from the pretrained checkpoint is far more sample-efficient.

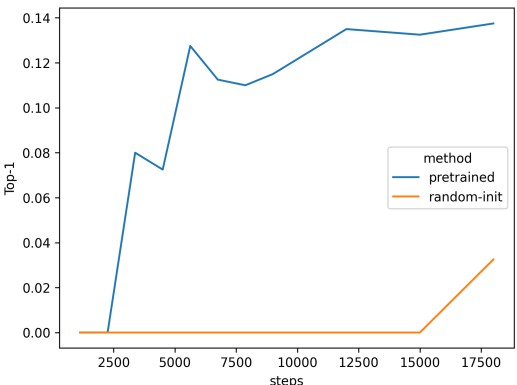

Figure 6: Comparing validation performance of a base model trained from a pretrained checkpoint to one trained from a randomly initialized checkpoint. One epoch of training is 9375 steps.

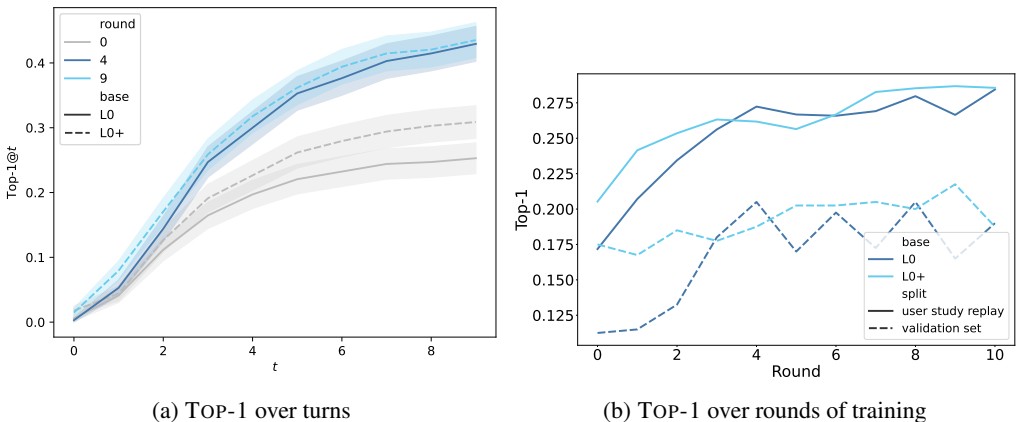

(a) TOP-1 over turns

(b) TOP-1 over rounds of training

Figure 7: Comparison between the L0 model from the original experiments and the L0+ model trained for longer. Round 0 corresponds to the LITERAL model.

## C    PRAGMATIC MODEL TRAINING

---

**Algorithm 1** Outer training loop

---

1: **procedure** TRAINING($L_{\theta_0}, S_{\phi_0}, \mathcal{H}, \mathcal{V}$, METRIC)
2:     **Input:** Base listener model $L_{\theta_0}$
3:     **Input:** Base speaker model $S_{\phi_0}$
4:     **Input:** Set of hypotheses $\mathcal{H}$
5:     **Input:** Validation pairs $\mathcal{V}$
6:     **Input:** Validation metric METRIC
7:     **Output:** Trained listener model $L_\theta$
8:     **for** $r = 0$ **to** $R_{\max}$ **do**
9:         Sample set of $k$ hypotheses $H \sim \mathcal{H}$
10:        $\mathcal{D}_r \leftarrow \{(h, \text{GENERATEPRAGMATICSPECIFICATION}(h)) \mid h \in H\}$
11:        $L_{\theta_{r+1}} \leftarrow \text{TRAINLISTENER}(L_{\theta_0}, \bigcup_{j=0}^{r} \mathcal{D}_j)$
12:        $S_{\phi_{r+1}} \leftarrow \text{TRAINSPEAKER}(S_{\phi_0}, \bigcup_{j=0}^{r} \mathcal{D}_j)$
13:        $\mathcal{H} \leftarrow \mathcal{H} \setminus H$
14:    **end for**
15:    **return** $\arg\max_r \text{METRIC}(L_{\theta_r}, \mathcal{V})$
16: **end procedure**

---

Finally, we finetune the the base models we train by iteratively using the speaker and listener models to generate data, and finetuning on the generated data using the algorithm shown above.

To generate the candidates at a particular round $r$, we use both the base models and the models from round $r$. Sampling from multiple models with different biases can help us draw a more diverse sample of programs and examples, and get a better approximation of the consistency matrix to perform RSA reasoning over. We draw 250 samples from each of the models we are sampling – the base listener, the listener from round $r$, the base speaker, and the speaker from round $r$.

As we note in the algorithm, we update the models by training from the initial parameters (of the base speaker or listener model) on all the datasets generated up to and including that round. We train for for $R_{max} = 20$ rounds with $k = 1024$ programs per round, generating specifications with up to $N_{utterances} = 10$ examples. We choose the model trained for 4 rounds based on the TOP-1 metric on the validation set of 40 examples. At each round we update the models using the AdamW optimizer for 1 epoch, using a learning rate of $5 \times 10^{-5}$. The batch size is set to 32.

## D    DATASET COLLECTION

We recruited 18 computer science graduate students. The data collection task proceeded in two phases – specification creation and specification verification. Some participants did the both tasks, while some only did the verification task. Participants who did both tasks spent approximately 3 hours on the task and were compensated with \$45, while those who did the verification task spent approximately 2 hours and were compensated with \$30. Participants were allowed to perform the task online at a location of their choice, and were allowed to perform the task across multiple sittings.

Each participant was given a short tutorial on the syntax and semantics of regular expressions, and small quiz where they had to enter three positive and three negative examples for a given regular expression to proceed to the rest of the study. Participants were then told about a communication game, and to provide examples that they would use if they were asking another person to write the regular expression for them – as they might on an online forum like StackOverflow.

Participants then went to the annotation screen for the specification creation task (if they took part, if not they proceeded directly to the next stage), where they were shown a regular expression and asked to enter examples consistent with it, in the interface shown in Figure 8. Consistency was automatically checked, so participants could not enter inconsistent examples. A minimum number of examples between 5 and 7 was randomly picked for each instance (to allow the participants from simply providing the same minimum number for each program), but no limit on the number of

---

**Algorithm 2** Approximate RSA inference

---

1: **procedure** GENERATEPRAGMATICSPECIFICATION($L_\theta, S_\phi, h$)
2:     **Input:** Listener model $L_\theta$
3:     **Input:** Speaker model $S_\phi$
4:     **Input:** Target program $h$
5:     **Output:** Specification $s$
6:     $s \leftarrow [\,]$
7:     **for** $i = 1$ **to** $N_{\text{examples}}$ **do**
8:         ▷ Sample examples from $S_\phi$ conditioned on $s$ and filter for consistency with $h$
9:         $E \leftarrow$ GETEXAMPLECANDIDATES($S_\phi, h, s$)
10:       ▷ Sample programs from $L_\theta$ conditioned on $s$ and filter for consistency with $s$
11:       $D \leftarrow$ GETDISTRACTORS($L_\theta, s$)
12:       **for** $p$ **in** $D \cup \{h\}$ **do**
13:         **for** $e$ **in** $E$ **do**
14:           ▷ Populate consistency matrix with whether $p$ is consistent with $e$
15:           $M[e, p] \leftarrow \mathbf{1}[p \vdash e]$
16:         **end for**
17:       **end for**
18:       ▷ Compute the literal listener distribution over the sample $M$ of the consistency matrix
19:       **for** $p$ **in** $D \cup \{h\}$ **do**
20:         **for** $e$ **in** $E$ **do**
21:           $L_{\text{literal}}[e, p] \leftarrow \frac{M[e,p]}{\sum_{p'} M[e,p']}$
22:         **end for**
23:       **end for**
24:       ▷ Populate consistency matrix with whether $p$ is consistent with $e$
25:       **for** $p$ **in** $D \cup \{h\}$ **do**
26:         **for** $e$ **in** $E$ **do**
27:           $S_{\text{RSA}}[e, p] \leftarrow \frac{L_{\text{literal}}[e,p]}{\sum_{e'} L_{\text{literal}}[e',p]}$
28:         **end for**
29:       **end for**
30:       $e^* \leftarrow \arg\max_e S_{\text{RSA}}[e, h]$
31:       $s \leftarrow s \oplus [e^*]$
32:     **end for**
33:     **return** $s$
34: **end procedure**

---

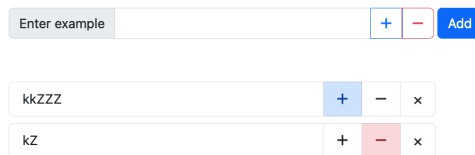

| Regular expressions from examples | User: dev | Regex cheatsheet |

### You are now the describer

Here, you are presented with a regular expression. You need to provide a set of examples that *describes* this regular expression to a guesser. Enter at least 5 examples.

### Regex: k{2,}Z{2,3}

Enter example     | + | − | Add

kkZZZ     +   −   ×
kZ     +   −   ×

Figure 8: Interface for providing examples for a regular expression

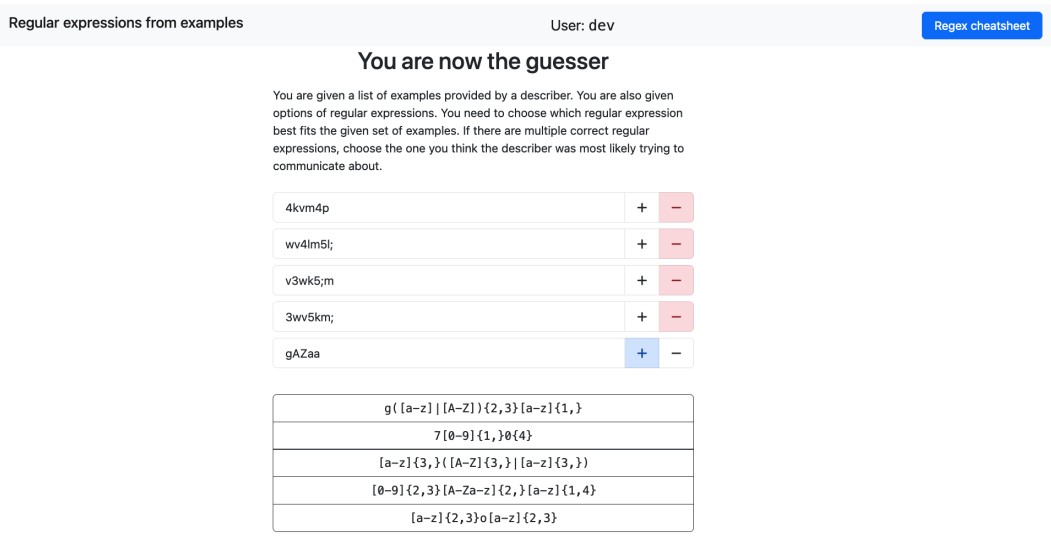

Figure 9: Interface for verifying a specification written by another human

examples was placed. Participants created specifications for 40 programs before proceeding to the verification task.

For the verification task, participants are shown a specification and 5 regular expressions, of which the target regular expression (that the specification was written for) is one, as shown in Figure 9. The others are distractors. To ensure the distractors are not too easy to distinguish from the target, we sample distractors using the LITERAL model. Of the 4 distractors, we attempt to find 2 which are consistent a subset of the specification, and 2 which are inconsistent with the specification. Since providing the entire specification might lead to too few consistent programs remaining, or distractors that are too hard to tell apart from the target, we use only the first example of the specification as input to the LITERAL model to generate distractors. Depending on whether they did the specification creation task, participants verify 40-90 specifications.

The dataset contains 440 examples in total. Of these, 423 have 2 verifications and 17 have one verification. Of the 440, 352 (80%) have two verifications that recover the target. 406 (92%) of the 440 have at least one verification recovering the target. We use 40 of the programs with two verifications recovering the target as the validation set.

# E   USER STUDY

To conduct the user study, we recruited 11 computer science graduate students. Participants spent approximately 1.5 hours on the task and were compensated with $25. Participants were allowed to perform the task online at a location of their choice, and were allowed to perform the task across multiple sittings. Figure 10 shows the UI.

Each participant was given a short tutorial on the syntax and semantics of regular expressions, and small quiz where they had to enter three positive and three negative examples for a given regular expression to proceed to the rest of the study. Participants were then told about a communication game, and informed that they would be playing the role of the speaker in the game and describing a target program. Each participant interacted with the three models in order – PRAX, LITERAL, and HFT – and cycled through the list in order, but first model they encountered was chosen randomly. Each model was assigned a color, and the participants were told the color of the model they were interacting with, but not given any other details about it. Participants provided examples one at a time, and observed the synthesizer's highest ranked guess at every step. If the guess matched the target program, the interaction ended. Participants were given the option to move to the next task after providing 10 examples. Each participant completed 31 instances they provided examples for

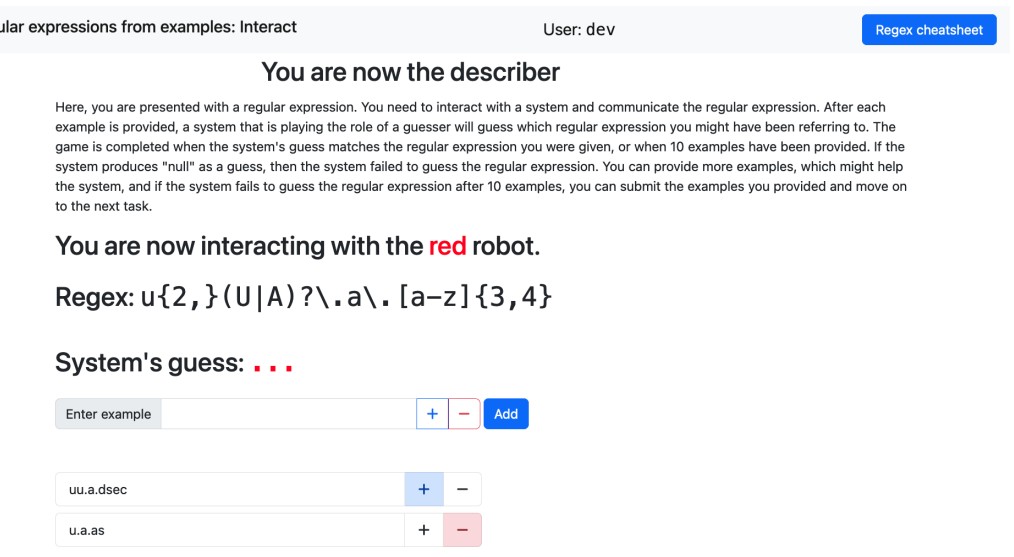

Figure 10: Screenshot of the user study UI with an example interaction

31 different programs, with one of the interactions from one of the particpicipants being excluded due to an error.

## F    PROMPTING AN LLM

We prompt the `gpt-3.5-turbo-instruct` variant of GPT-3.5. We provide examples of 5 program-specification pairs in the context of a code snippet, followed by a set of examples. We sample 10 generations with temperature $t = 0.7$ and $p = 0.9$. We sample for a maximum of 64 tokens, or stop tokens that indicate that the regular expression is complete are sampled.

We rank the generated completions using the sum of token log probabilities, and choose the highest scoring program consistent with the given answer as the top-1 guess, and other consistent programs as the top-10 guesses to calculate the TOP-1 and TOP-10 metrics.

## G    SPEAKER QUALITY

One measure of the speaker's quality is the loss evaluated in examples provided by humans. Evaluating loss allows us to see if human provided examples are more likely under a model's distribution, without penalizing the model for the exact example not appearing in a set of examples decoded from the model (since there are many possible examples that are consistent with a program, and multiple possible informative examples too). We see in Figure 11 that even without being trained on any human-provided examples, the speaker loss on human-provided validation examples reduces.

## H    CONNECTIONS TO ITERATED BOOTSTRAPPED TRAINING

Methods such as DreamCoder (Ellis et al., 2021) and Memoized Wake Sleep (Hewitt et al., 2020) use iterated bootstrap training to build a hierarchy of abstractions to solve tasks more effectively over iterations of training. At each iteration of training, they create tasks by sampling a program from a changing library of abstractions, and executing the program on a fixed distribution of inputs to train a synthesizer to work with updated abstractions.

Our method on the other hand is aimed at allowing a synthesizer to reason about how examples are chosen to demonstrate programs from a fixed library. We change the distribution of examples being drawn from a trained speaker model over the course of iterated bootstrapped training towards more informative examples.

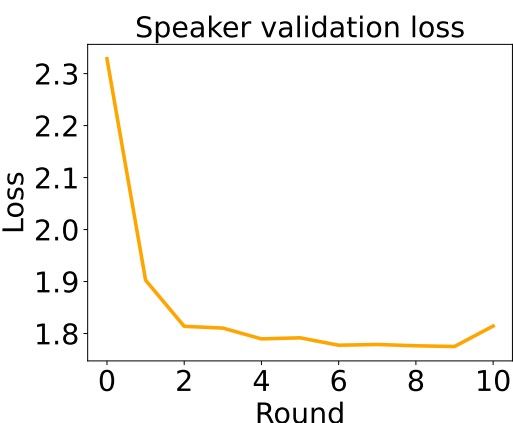

Figure 11: Speaker loss on the validation set computed over rounds of training

