# OpenReview forum: "Generating Pragmatic Examples to Train Neural Program Synthesizers"
_ICLR.cc/2024/Conference — ICLR 2024 poster_

### Official Review · Reviewer_3Xjo · 2023-10-22

**Soundness:** 2 fair
**Presentation:** 3 good
**Contribution:** 3 good
**Rating:** 6
**Confidence:** 2

**Summary:**

The authors present a framework built on the Rational Speech Acts model that iteratively tunes a listener and speaker model to generate programs fitting a spec. This framework can essentially be viewed as a bootstrapping method to build a dataset and listener and speaker models in a more efficient way than blindly sampling programs. Experiments on a regex dataset show that this framework outperforms naively training a single L/S model pair with a moderately sized dataset as well as prompting a LLM. This framework also outperforms using a human dataset instead of a listener.

**Strengths:**

- The presented method outperforms the literal and GPT baselines. It also outperforms HFT which suggests that the iterative listener training makes a difference.
- The method outperforms a small human labeled dataset, which suggests that it may be better to use this method if one does not have access to lots of human annotations.
- The method converges at a comprable rate to the literal and HFT methods, so the bootstrapping method appears to work.

**Weaknesses:**

- The method is only tested on a regex dataset. This ignores programs that cannot be written as regexes and more complicated programs.
- This paper only compares the pragmatic framework against the literal and HFT baselines, which are derivatives of the pragmatic framework. While there is a comparison against GPT3.5, I would have liked to see other program synthesis baselines that do similar things for a more thorough comparison.
- There are no ablations on the number of samples needed to get good performance and other similar hyperparameters.

**Questions:**

- Will the dataset be released?
- How do you guarantee your base program/sample dataset has sufficient coverage to get a usable listener/speaker model? Do you have a prior over which programs and samples may be more useful for training a model?
- What distribution do you use to sample the set of rows and columns to update for RSA?

---

> ### Author Response · Authors · 2023-11-16
> **Response to Reviewer 3Xjo**
>
> > Will the dataset be released?
>
> Yes! It’s included in the supplement.
>
> > How do you guarantee your base program/sample dataset has sufficient coverage to get a usable listener/speaker model?
>
> We train the base models on randomly sampled program-specification pairs to initialize them as capable program synthesis models as described in section 3.2 of the paper and section 1 of the General Response.
>
> > Do you have a prior over which programs and samples may be more useful for training a model?
>
> Yes, we use a prior over programs based on Ye et al. (2020) that is described in section 4.1 and appendix A of the paper.
>
> > What distribution do you use to sample the set of rows and columns to update for RSA?
>
> We use the listener model to generate programs consistent with the prior example (“ab”, ✓) and the speaker model to generate examples consistent with the target program +b* as shown in Figure 1 of the paper. So, for the given example, in the second turn of the simulated interaction we may get a set of programs [a?b?, a+b+, a*b+c?] and examples [(“aab”, ✓), (“abb”, ✓), (“aa”, ✓), (“b”, ✗)], and use those to construct the consistency matrix over which we perform RSA inference.
>
> > This paper only compares the pragmatic framework against the literal and HFT baselines, which are derivatives of the pragmatic framework. While there is a comparison against GPT3.5, I would have liked to see other program synthesis baselines that do similar things for a more thorough comparison.
>
> In section 1 of the General Response, we detail why the Literal model is not derivative of our pragmatic approach, and in itself a standard program synthesis baseline. In section 3 we describe some findings from running a symbolic regex-specific program synthesizer.
>
> ## Ablations
> > There are no ablations on the number of samples needed to get good performance and other similar hyperparameters.
>
> Figure 5 presents the performance of the model over rounds of training. Since each round corresponds to training on a new set of (in this case 1024) program-specification pairs, this is an ablation that looks at the number of samples (where each program-specification pair counts as a sample) required to get good performance. We find that model performance increases over rounds of training, highlighting the value of iterated training.
>
> We have also added a comparison to one epoch of training on two different sets of program-specification pairs. One set of 5120 pairs is generated using the base speaker and listener models, corresponding to training on the same number of programs as the Pragmatic model, but in one round, effectively ablating the effect of iterated training. Training in a single round does not perform as well as training in multiple rounds of training.
>
> We also compare to training on 400 pairs from the speaker in the 5th round of training, and find that this performs quite well, suggesting that a speaker in later rounds of training produces informative examples.
>
> If you were referring to a different ablation, please let us know so we can clarify our method.
>
> ## References
> Xi Ye, Qiaochu Chen, Isil Dillig, and Greg Durrett. Benchmarking multimodal regex synthesis with
> complex structures. In Proceedings of the 58th Annual Meeting of the Association for Computa-
> tional Linguistics, pp. 6081–6094, Online, July 2020. Association for Computational Linguistics.
> doi: 10.18653/v1/2020.acl-main.541. URL https://aclanthology.org/2020.acl-main.541.

---

> > ### Author Response · Authors · 2023-11-20
> > **Update**
> >
> > Thanks Reviewer 3Xjo for your thorough review! We have added some more information to our response, and we hope you will  take a look and consider updating your score.

---

> > > ### Comment · Reviewer_3Xjo · 2023-11-20
> > >
> > > >We train the base models on randomly sampled program-specification pairs to initialize them as capable program synthesis models as described in section 3.2 of the paper and section 1 of the General Response.
> > >
> > > I re-read those sections and my understanding is the same as before in that you sample uniformly from the input space. As this input space can be very large, how do you ensure that you get sufficient signal for "rare" programs?

---

> > > > ### Author Response · Authors · 2023-11-20
> > > >
> > > > In program synthesis from examples (like RobustFill and DeepCoder) programs are sampled from a prior distribution, without special treatment to “rare” programs. the reasoning being once the synthesizer has been expoed to a sufficient number of programs in this prior, it should generalize to “rare” programs at inference time. We follow that to build our base model by naively sampling the input space (50% positive, 50% negative examples). The strong baseline performance of the literal synthesizer indicates that this is quite effective.
> > > >
> > > > However, we share your insight that providing more targeted examples can aid learning — this is precisely the purpose of our proposed method of choosing pragmatic examples, and we demonstrate that this is indeed effective. Our work is orthogonal to this concern, as we build our pragmatic synthesizer “on top of” this base synthesizer. Thus, we simply followed the conventions of prior works in training the base synthesizer. A pragmatic synthesizer that is built on a stronger base model should be stronger, since it will result in a higher quality sample of the consistency matrix for RSA.

---

> > > > > ### Comment · Reviewer_3Xjo · 2023-11-20
> > > > >
> > > > > Thank you for your response. I have raised my score to a 6.

---

> > > > > > ### Author Response · Authors · 2023-11-20
> > > > > >
> > > > > > Thanks for your consideration of our response and engaging to update your judgement!

---

### Official Review · Reviewer_h5RG · 2023-10-30

**Soundness:** 4 excellent
**Presentation:** 3 good
**Contribution:** 4 excellent
**Rating:** 8
**Confidence:** 4

**Summary:**

In this paper, the authors propose a method for program synthesis (specifically the programming-by-examples variety of it) framed as a two-player game. A _speaker_ model $S$ is charged with generating informative examples, which a _listener_ model $L$ uses to generate programs; within this setup, a given set of examples can be consistent with multiple programs. The authors get around this difficulty by devising an approximation of a pre-existing Bayesian scheme, dubbed RSA. Using LLMs as speaker and listener models, and by performing RSA only on a partial consistency matrix $M$ of programs/examples pairs sampled from them (instead of the full matrix which is prescribed by exact RSA), the authors make it possible to select the examples which are most informative for a given program, resolving the aforementioned ambiguity. These examples are then added to the dataset of (example, program) pairs and used to train the speaker and listener in an Expert Iteration fashion.
The authors demonstrate that their proposed method, trained using only synthetic data and their ExpIt-like procedure, performs better than a set of baselines, including a model trained on high quality data sourced from human annotators, and GPT 3.5.

**Strengths:**

- While RSA is not novel in itself, the paper is anyway quite novel in making it applicable to a large-scale program synthesis task, and in using LLMs to model speaker and listener.
- The paper is accessible and written well, though it does contain a very large number of details and some typos.
- The authors set up a comprehensive experimental pipeline inclusive of humans in the loop. Most papers on code synthesis do not actually test their methods "in the wild", while this paper does.
- The proposed method does beat the considered baselines by a good margin.

**Weaknesses:**

- It appears that one assumption of RSA is that given a certain example, the literal listener $L_0$ should assign equal probability to all programs consistent with it. This is unlikely to be the case once $L_0$ is a neural net, unless this has been perfectly trained.
- Synthetic data needs to be available to pre-train the literal speaker and listener models. These might not be available when considering more rich "programming" languages than regexs.
- Both the training and evaluation protocols are quite complex, meaning that multiple reads of the paper  are necessary to get all of the details.
- The set of baselines considered is somewhat narrow, and does not include any previous efforts on the particular task considered (i.e. inferring regular expressions). The authors compare only against effectively an ablation of their own method (LITERAL), their method but trained on a set of human-annotated data (HFT), and a generalist LLM (GTP 3.5).

**Questions:**

- As mentioned above, the RSA framework assumes $L_0$ to assign equal probability to all consistent programs, which is not going to be the case once it is approximated with a Neural Net. Could the authors comment on this?
- The literal listener $S_0$ is not defined in section 2. Only $L_0$ and $S_1$ are defined. Could the authors provide a definition?
- Why didn't the authors include any baselines (neural or not) specific to the task of inferring regexs?
- Some details of the evaluation protocol are a bit obscure. First of all, how do the authors compute their top-1 metric on the validation dataset, when doing model selection? They only provide a definition of the metric as part of the final human trial, so it's not clear how the model would be prompted when computing it at the model selection stage.
- The authors state that "TOP-1@$t$ measures whether the model's top-1 matches intended regular expression at any point at turn $t$ of the interaction". Is it at any point, or a turn $t$? Since the interaction ends after a match is achieved, it can only be at turn $t$, but the text is ambiguous.
- The authors detail their inference procedure in section 4.4. Therein they state that programs inconsistent with a given example are filtered out. Shouldn't they be left in in order to build the consistency matrix $M$? Does this paragraph only detail the inference protocol for the evaluation phase, or also for the training phase? This is not clear.
- I assume that the numbers reported in the tables refer to the _fraction_ of instances in which an interaction was successful at time $t$, so it's technically incorrect that the metrics measure "whether" something happens. They actually measure "how often" it happens over a set of interactions. It would be helpful to the reader to amend this somewhat inaccurate language.
- From section 4.7: "4 shows the progression of..." 4 what? I assume that this actually refers to figure 3 and this is a typo.
- From the intro: "A synthesizer trainer in the style of Devlin et al...." what does this mean? Could the authors be more explicit?

**Details Of Ethics Concerns:**

No concerns.

---

> ### Author Response · Authors · 2023-11-16
> **Response to Reviewer h5RG**
>
> > As mentioned above, the RSA framework assumes to assign equal probability to all consistent programs, which is not going to be the case once it is approximated with a Neural Net. Could the authors comment on this?
>
> This is correct – it is not possible to model a uniform distribution over all consistent programs using the literal listener. We also added a prior term to the description of the L0 computation, thanks for pointing it out. This prior is implicitly modeled by the base model when it is trained. However, this non-uniform prior is used only when we sample from the base model. We follow prior work (Pu et al., 2020) and explicitly impose a uniform prior over the sampled programs (that we use to populate the consistency matrix we perform RSA inference over). We have amended the text of Section 3.2 to convey this more clearly.
>
> > The literal [speaker] $S_0$ is not defined in section 2. Only $L_0$ and $S_1$ are defined. Could the authors provide a definition?
>
> We amended the paper to include the definition of $S_0$ analogous to $L_0$: $S_0(example|program) \propto M(example, program)P(example)$
>
> > Why didn't the authors include any baselines (neural or not) specific to the task of inferring regexs?
>
> In section 3 of the General Response we describe some findings from running a symbolic regex-specific program synthesizer.
>
> > Some details of the evaluation protocol are a bit obscure. First of all, how do the authors compute their top-1 metric on the validation dataset, when doing model selection? They only provide a definition of the metric as part of the final human trial, so it's not clear how the model would be prompted when computing it at the model selection stage.
>
> Here is how we perform model selection. Given a model, we present the examples (a single string-label pair) from the validation set one example at a time, in sequence, and count the fraction of instances in which the model recovers the correct program. This fraction is used to score the models. This is similar to the replay setting which we use to evaluate GPT-3.5.
>
> > The authors state that "TOP-1@t measures whether the model's top-1 matches intended regular expression at any point at turn of the interaction". Is it at any point, or a turn t? Since the interaction ends after a match is achieved, it can only be at turn , but the text is ambiguous.
>
> Yes, this is a typo, thanks for pointing it out! We mean that for a turn t, the TOP-1@t metric measures the fraction of tasks solved after t turns are complete (that is, the user has provided t examples).
>
> > The authors detail their inference procedure in section 4.4. Therein they state that programs inconsistent with a given example are filtered out. Shouldn't they be left in in order to build the consistency matrix M? Does this paragraph only detail the inference protocol for the evaluation phase, or also for the training phase? This is not clear.
>
> The protocol is for both training and inference. We want the model to learn how to disambiguate between consistent programs, assuming that the base model is already effective at finding a consistent program. Note that the programs sampled are consistent with the given specification, but may be inconsistent with other examples sampled by the speaker that are not yet added to the specification, which builds a consistency matrix with both 1s and 0s.
>
> > I assume that the numbers reported in the tables refer to the fraction of instances in which an interaction was successful at time, so it's technically incorrect that the metrics measure "whether" something happens. They actually measure "how often" it happens over a set of interactions. It would be helpful to the reader to amend this somewhat inaccurate language.
>
> We will amend the explanation of the method to make this clear.
>
> > From section 4.7: "4 shows the progression of..." 4 what? I assume that this actually refers to figure 3 and this is a typo.
>
> This is a typo, thanks for catching it!
>
> > From the intro: "A synthesizer trainer in the style of Devlin et al...." what does this mean? Could the authors be more explicit?
>
> We elaborate on this in section 1 of the General Response. We have also updated section 3.2 to make this more clear.

---

> > ### Comment · Reviewer_h5RG · 2023-11-20
> >
> > I thank the authors for their response to my questions and the modifications they have carried out on the paper.
> > I do not have any further points to raise and those I did raise were anyway fairly minor in nature. My score remains therefore unchanged.

---

> > > ### Author Response · Authors · 2023-11-22
> > >
> > > Thank you for considering our response!

---

### Official Review · Reviewer_Lieb · 2023-10-31

**Soundness:** 3 good
**Presentation:** 3 good
**Contribution:** 3 good
**Rating:** 6
**Confidence:** 3

**Summary:**

The paper scales pragmatic inference for program synthesis to realistic problem
sizes using neural networks. The authors introduce a listener and a speaker
neural model and train these iteratively. The models are used to generate
datasets containing increasingly informative program specifications, and they
themselves are trained further on these datasets in each iteration. To build the
dataset, the models suggest candidate specifications, and the specifications to
be included are chosen from these using pragmatic inference (the Rational Speech
Acts framework).

The method is evaluated on the task of inferring regular expressions from a set
of examples in a human interaction study with 11 participants and outperforms a
base literal model, a human finetuned model, and GPT-3.5.

**Strengths:**

Scaling up pragmatic inference for realistic program synthesis could open up
promising future research.

The paper is clearly written with illustrative figures. I found the explanation
of the pragmatic model of program synthesis really good and easy to follow.

The paper includes a real-world study of program synthesis with 11 human
participants.

**Weaknesses:**

I believe the main weakness of the paper is that the presented method is not
compared to any existing neural program synthesis system (like DeepCoder,
PCCoder, DreamCoder, CrossBeam, LambdaBeam, etc.). I think this would be
important as the main thesis of the paper is scaling up pragmatic inference to
the level of these systems. It would also be good to include a domain for
synthesizing programs that's more general and widespread in the literature than
regexes.

I think that Section 4.6 about the human annotated dataset should be earlier as
it's already referred to earlier.

Some typos:
- Introduction: "an user", "coorporative", "human ... atempt to communicate",
  "of of"
- page 3 top line "an sampled example"
- 4.3 Measurement: "top-1 matches THE intended regular expression"
- 4.7 Results: "informatively", "4" should be "Figure 3"
- 6 Related work: "datas"

**Questions:**

I couldn't understand the argument for sampling a subset of the consistency
matrix $M$ in 3.3. If $M$ is sparse, why does that allow us to sample a subset
of the rows and columns? Wouldn't we mostly sample zeros? Or is there a strategy
for sampling (e.g., sampling dense areas) which is not mentioned?

I also don't understand exactly how the conditioning on previous examples are
done when sampling in terms of the consistency matrix. Could you elaborate on
that?

---

> ### Author Response · Authors · 2023-11-16
> **Response to Reviewer Lieb**
>
> ## Sampling from the consistency matrix
> > I couldn't understand the argument for sampling a subset of the consistency matrix in 3.3. If M is sparse, why does that allow us to sample a subset of the rows and columns? Wouldn't we mostly sample zeros?
>
> We use M to refer to the _entire_ consistency matrix with all programs and examples. So, sampling a set of columns at random would mean choosing a set of random programs, and sampling a set of rows would be choosing a set of examples at random. In the context of Figure 1 (where the target program is a+b*), an example of these samples may be [“p+q*”, “hg*”, “jk+”], and [(“xyz”, ✓), (“ij”, ✓), (“jld”, ✓)]. The part of the consistency matrix corresponding to these examples would be full of 0s.
> Instead, if we use the listener to generate programs consistent with the prior example (“ab”, ✓) and the speaker to generate examples consistent with the target program +b*, we may get a set of programs [a?b?, a+b+, a*b+c?] and examples [(“aab”, ✓), (“abb”, ✓), (“aa”, ✓), (“b”, ✗)], as shown in the figure. We can see again in the figure that this matrix is a lot less sparse than the case of choosing programs and examples to populate the matrix as discussed earlier.
>
> > Or is there a strategy for sampling (e.g., sampling dense areas) which is not mentioned?
>
> The only strategy we use is filtering for consistency with the previous examples (for the listener) and the program (for the speaker). We do not use any other sampling strategies.
>
> ## Conditioning on previous examples
> > I also don't understand exactly how the conditioning on previous examples are done when sampling in terms of the consistency matrix. Could you elaborate on that?
>
> When generating examples at each turn, we resample the rows (programs) and columns (examples) based on the examples already added to the specification. This allows us to look at a relevant sample of the consistency matrix, as explained above.
> Referring to Figure 1 of the paper, we unroll an interaction between the speaker and listener models over a sequence of turns. In the first turn, there are no prior examples. The speaker generates examples only based on the target program (in this case a+b*), and the listener synthesizes programs that satisfy an empty specification. We then perform RSA inference and select an example to add to the specification. In the figure, this example is shown to be (“ab”, ✓). For the next turn, we use the specification built so far (now [(“ab”, ✓)]) as inputs to the speaker and listener models, and sample the next example as shown in the figure. This example is added to the specification, and the process is repeated.

---

> > ### Author Response · Authors · 2023-11-20
> > **Update**
> >
> > Thanks Reviewer Lieb for your thorough review! We have added some more information to our response, and we hope you will  take a look and consider updating your score.

---

> > > ### Comment · Reviewer_Lieb · 2023-11-21
> > > **Response to authors**
> > >
> > > Thank you for your thoughtful answers and clear explanations.
> > >
> > > Especially point 2 in the general response - that the proposed method is model agnostic - improved my understanding, I have not thought of it this way. I've increased my score because of this, and I think this could be emphasized in the paper.
> > >
> > > I still think that it would be good to have a domain besides regexes. Also, if I understand correctly, the additional experiments have a mismatch between domain and method (as not all of the constructs can be used).

---

> > > > ### Author Response · Authors · 2023-11-22
> > > >
> > > > Thank you for engaging and updating your judgement!
> > > >
> > > > We have made some edits to emphasize that our model isn’t dependent on this specific neural parameterization, and we will also explore ways we can make this more clear for the camera ready.
> > > >
> > > > The additional experiments do not reveal a mismatch between domain and method, since the difference is in distribution and not expressibility. Note that numeric ranges are expressible by these (for example, `A{2,4}` is semantically equivalent to `AAA?A?` ) but the design of these synthesizers forces these constructions to be ranked lower and thus harder to find. These methods could be tailored to express these constructs using the abstractions we do in our DSL, however that would require significant time and effort.

---

### Official Review · Reviewer_Dg1v · 2023-10-31

**Soundness:** 3 good
**Presentation:** 3 good
**Contribution:** 3 good
**Rating:** 6
**Confidence:** 3

**Summary:**

This paper focuses on program synthesis by example for regexes. It aims to learn a program synthesis model that reasons pragmatically. As many possible programs can meet ambiguous input example specifications, counterfactual thinking should be usefully employed to differentiate among the many valid hypotheses. Other works have investigated this possibility, most under the rational speak acts (RSA) framework, but this kind of reasoning is intractable to do exactly for non-trivial domains. The paper suggests a bootstrapped learning approach to overcome this limitation, by jointly learning a speaker model (which suggests pragmatic examples given an input program) and a listener model (which suggests a likely program, given a list of assumed pragmatic examples). Over multiple rounds, these networks are trained on one another’s predictions, chosen according to an RSA methodology, made tractable by restricting the hypothesis space according to the programs sampled from the model. Experiments with human-trials demonstrate that listener models trained in this framework perform better than comparison approaches.

**Strengths:**

I enjoyed this paper and I would support its acceptance into the conference proceedings.

The proposed approach is sensible and well-explained. The methodology appears quite general, and should be able to be used broadly as it requires no human GT data during training. In fact, the human GT data that is used in validation seems like it could be removed, as based on the trend in figure 5 it doesn’t appear as though the method is overfitting in any sense over bootstrapping rounds.

The experimental design and presentation is sound and convincing for the regexes inference domain. The paper mainly validates the proposed system with human-trials, which makes sense as its hard otherwise to source ‘pragmatic’ examples, and it confirms the system would actually be easier to work with for an end-user

**Weaknesses:**

While a lot of effort has gone into validating the system is working for this particular regex domain, the paper does not explore other problem settings to any degree. I don’t think this is a major limitation, but it is probably holding the paper’s rating back slightly. The impressive results on one-domain are likely of interest to a subset of the ICLR community, but showing that this methodology can generalize effectively across domains would broaden this interest. As a note, I don’t even think these other domains would need to be *more difficult* than regexes (e.g. like python code generation mentioned in the conclusion), but could even be other domains of similar complexity.

In some ways the comparison against HFT is a bit unfair, as the proposed method has effectively unlimited training data (although only a set amount of bootstrapping rounds are employed), whereas HFT is fine-tuned with a fixed amount of human-feedback. To get a “fairer” upper-bound of how “good” the pragmatic examples produced by the system are with respect to the human provided exemplars, it might be good to include an additional condition where the listener model is fine-tuned on a fixed amount of data (i.e. the same amount as used in HFT) where the I/O examples are produced by the final speaker model.

Minor:

There is a connection to be made between the proposed method and bootstrapped “wake-sleep” approaches for program synthesis [1,2]. Both learn “generative” and “inference” models that learn on one another’s outputs. The modeling set-ups are different, as these wake-sleep approaches move towards a target distribution, whereas the proposed method optimizes for synthetic training data that matches a prior desiderata (pragmatic I/O examples), but these ideas are close enough that they should be discussed within the related work section.

[1] DreamCoder: growing generalizable, interpretable knowledge with wake–sleep Bayesian program learning
[2] Learning to learn generative programs with Memoised Wake-Sleep

**Questions:**

(1) The proposed system effectively improves the listener model by finding “better” synthetic training I/O examples, where better means there is a pragmatic connection between the examples and the target program. However, it’s unclear if this improvement changes the upper-bound of the listener model performance, or if it just helps the listener model reach a good performance with less training iterations. Training the base model for 300k programs, for a single epoch, it's not clear whether the model has started to plateau in performance. It would be helpful to provide evidence that the base model has saturated, by e.g. plotting validation performance over pretraining iterations. What would this plot look like?

(2) It’s also not clear to me why starting from a pretrained model like ByT5 would be necessary or helpful. The programs come from a constrained DSL, where unlimited synthetic data can be sampled, so it should be possible to train the base models from scratch. It would be good to include an ablation on how starting with or without pretraining affects the proposed method. What is the justification for starting with a pretrained model?

(3) Much of the evaluation is based off of human-interactions, which is present only in limited quantities. Are there any metrics which could be evaluated without human interactions? For instance, what about the following set-up:

1. Pick a target regular expression and a single I/O example at random
2. The listener samples a target expression given the current specification
3. With respect to (2), the speaker samples an example, which is annotated as consistent/inconsistent by an oracle
4. Repeat steps 2 and 3 until the listener samples the *correct* target expression

The metric would then be the number of example generations needed for the listener to predict the correct regular expression, where the idea would be that as the speaker is better at producing "pragmatic" examples it should require less steps versus a baseline that for instance randomly sampled an example. Beyond serving an evaluation set-up that requires no human-data, this kind of framework could conceivably even be useful for “real-world” applications: e.g. this could reduce the burden on an end-user, who instead of having to think up new examples as input, would just need to label them.

## Minor:

(4) Is the hypotheses set in Appendix D randomly sampled programs from the DSL? Please make this clear. More generally, some of the terminology used in the pseudo-code could be more directly mapped back to concepts in the main paper, or given a more detailed treatment in the supplemental text.

There is a typo in the figure 3 caption: metrix

---

> ### Author Response · Authors · 2023-11-16
> **Response to Reviewer Dg1v**
>
> ## 1. Speaker generates high-quality examples in later rounds of training
> > In some ways the comparison against HFT is a bit unfair, as the proposed method has effectively unlimited training data (although only a set amount of bootstrapping rounds are employed), whereas HFT is fine-tuned with a fixed amount of human-feedback. To get a “fairer” upper-bound of how “good” the pragmatic examples produced by the system are with respect to the human provided exemplars, it might be good to include an additional condition where the listener model is fine-tuned on a fixed amount of data (i.e. the same amount as used in HFT) where the I/O examples are produced by the final speaker model.
>
> We have added a comparison to Figure 5 in the updated draft.
>
> The ability to train on large amounts of synthetically generated data has been vital in making neural and neuro-symbolic program synthesis approaches (such as RobustFill, DeepCoder, etc.) work. Our work is in a similar spirit, and asks whether we can make such synthetically generated data more closely resemble the examples that a human user would provide while using the system.
>
> Hence, our proposed approach can be used to generate a very large number of examples at a much lower cost than having users provide examples. So, even if each example produced by our method is of lower quality or utility than a human-provided example, the ability to scale our approach better than having users annotate data means that we can achieve comparable results at a lower cost, even if we use more examples overall.
>
> We did run the experiment of finetuning the base model on 400 program-specification pairs from the speaker in the 5th round of training, and find that it performs quite close to the full pragmatic training, suggesting that these examples have comparable value to human-provided examples. Note that these 400 examples cannot be obtained without going through the other rounds of training too, and this experiment is meant to illustrate the utility of examples.
>
> ## 2. Base model performance plateaus during training
> > (1) The proposed system effectively improves the listener model by finding “better” synthetic training I/O examples, where better means there is a pragmatic connection between the examples and the target program. However, it’s unclear if this improvement changes the upper-bound of the listener model performance, or if it just helps the listener model reach a good performance with less training iterations. Training the base model for 300k programs, for a single epoch, it's not clear whether the model has started to plateau in performance. It would be helpful to provide evidence that the base model has saturated, by e.g. plotting validation performance over pretraining iterations. What would this plot look like?
>
> Figure 6 in the updated draft plots the validation metric (Top-1 success) on the validation set (40 program-specification pairs). We see that the base model has started to plateau in performance after 1 epoch of training (9375 steps). We have also updated Appendix B with this experiment.
>
> ## 3. Pretrained checkpoints help the base models
> > (2) It’s also not clear to me why starting from a pretrained model like ByT5 would be necessary or helpful. The programs come from a constrained DSL, where unlimited synthetic data can be sampled, so it should be possible to train the base models from scratch. It would be good to include an ablation on how starting with or without pretraining affects the proposed method. What is the justification for starting with a pretrained model?
>
> We also observe a substantial gain in sample-efficiency from using a pretrained model (as seen in the plot in Figure 6 of the updated draft). But, in general we agree with your statement that with the ability to generate unlimited synthetic data, we expect that we can start with a randomly initialized model and train until we are able to match the performance of starting with a pretrained model. However, since our goal here is to have a strong base synthesizer (while being agnostic to the specific type of base synthesizer), we opted to use a pretrained model in the interest of sample-efficiency.
>
> ## 4. Human vs automated evaluation
> > (3) Much of the evaluation is based off of human-interactions, which is present only in limited quantities. [...] would just need to label them.
>
> Since the goal of building this kind of synthesizer is to better interpret human communication, the most faithful evaluation is to have humans interact with this kind of system. Having a model stand in for a human in evaluation might not be representative of the results of human interaction. With specific reference to an algorithm like the one you proposed, the choice of a random I/O example at the first turn is already a significant deviation from human behavior, since a human’s first example would also be chosen informatively. This would in turn influence every future turn of interaction too.

---

> > ### Author Response · Authors · 2023-11-20
> > **Update**
> >
> > Thanks Reviewer Dg1v for your thorough review! We have added some more information to our response, and we hope you will  take a look and consider updating your score.

---

> ### Comment · Reviewer_Dg1v · 2023-11-20
>
> I appreciate the author's responses - I remain positive on the paper and I still supports its acceptance.
>
> ```We did run the experiment of finetuning the base model on 400 program-specification pairs from the speaker in the 5th round of training, and find that it performs quite close to the full pragmatic training, suggesting that these examples have comparable value to human-provided examples. Note that these 400 examples cannot be obtained without going through the other rounds of training too, and this experiment is meant to illustrate the utility of examples.```
>
> Thank you for running this experiment -- I agree with your assessment, and to put it on my own words: this experiments shows that speaker model (once trained) is able to generate training pairs at a similar quality to humans (in terms of impact on training). This is a compelling point in favor of the method that I think should make its way into the discussion.
>
>  ```Since the goal of building this kind of synthesizer is to better interpret human communication, the most faithful evaluation is to have humans interact with this kind of system. Having a model stand in for a human in evaluation might not be representative of the results of human interaction. With specific reference to an algorithm like the one you proposed, the choice of a random I/O example at the first turn is already a significant deviation from human behavior, since a human’s first example would also be chosen informatively. This would in turn influence every future turn of interaction too.```
>
> Yes certainly there is no perfect replacement for human-studies, and any automatic metric that tries to capture human-interactions will have flaws, but I still maintain that trying to develop such metrics can be quite useful, for both this work and for future works that will try to improve within this design space. If the choice of the first random I/O has a massive impact on how many tries it takes to find the *right expression*, this is something that can be quantified with this metric (e.g. by choosing the I/O pair randomly, or having a human provide only the first I/O pair).
>
> Figure 6 – not a major point, but I really think more epochs should be added to this plot, there is still very little evidence of plateauing, especially with how noisy top-1 is as a metric (as it doesn't take into account near misses).
>
> Finally, I would once again encourage the authors to consider discussing the connections to the bootstrapped learning methods [1] and [2] -- these are useful ideological connections that future readers of the paper may benefit from.

---

> ### Author Response · Authors · 2023-11-22
>
> Thank you for engaging without response!
>
> We re-ran the literal listener training for additional epochs, and found that there is an improvement in performance as the base model is trained for longer. Since we train our models with linear learning rate decay, we tried two values for the maximum number of epochs — 3 and 20 (the model being trained for 20 epochs did not finish training in the course of author response, but we include partial results from it), and we show the validation performance in Figure 7. We see that the model trained for 3 epochs achieves higher performance. To ensure a fair comparison of our pragmatic training procedure against a stronger listener, we also present preliminary results training the pragmatic model starting with the new base model. Given time constraints, these experiments did not finish running, but preliminary evidence already shows that our pragmatic training procedure is able to significantly outperform the stronger base model even before convergence (Figures 8a,b). We this conclude that, irrespective of the base model, our approach is able to improve the performance of a synthesizer trained with uninformatively selected examples. Please refer to the PDF for these figures.
>
> We have also added some discussion about other iterated bootstrap training to the paper to highlight these connections to the literature.

---

### Author Response · Authors · 2023-11-16
**General response**

General
We thank all the reviewers for their thoughtful and detailed feedback! We are glad that reviewers noted the
- effectiveness (Dg1v, Lieb, h5RG, 3Xjo)
- well-motivated nature and broad applicability (Dg1v)
- accessibility and clarity of presentation (Dg1v, Lieb, h5RG)
- real-world user study-based evaluation (Dg1v, Lieb, h5RG)

of our method.

However, a common concern raised is the lack of comparison to other program synthesis baselines. We address this concern in two ways:
We clarify what we believe to be a misunderstanding of the role of the Literal model in our experiments, which is itself a program synthesizer similar to RobustFill
We present our findings from running an additional symbolic synthesizer for the regular expression domain
We also address specific weaknesses pointed out by reviewers in individual responses to each reviewer.

# 1. RobustFill and the Literal synthesizer
Multiple reviewers seem to have understood the Literal model in our experiments as an ablation of our full Pragmatic model (h5RG “effectively an ablation of their own method”, 3Xjo “literal and HFT baselines, which are derivatives of the pragmatic framework”). However, we would like to clarify that the Literal model is an instantiation of a training paradigm that has been used to train strong neural program synthesizers such as RobustFill (Devlin et al., 2017) in prior work. We should have made this connection clearer, and thank the reviewers for raising this concern.

The RobustFill model is a deep neural network trained to map a set of randomly chosen input-output examples to a program. It is combined with an inference procedure that draws a sample of programs from the trained model conditioned on the given examples, and chooses the highest scoring consistent program. This is the Literal model in our experiments. Since it is trained on a set of uninformatively sampled input-output pairs, we designate this approach as the “Literal” model. The primary difference in our implementation of RobustFill is our use of a larger (pretrained) Transformer model, and a different sampling procedure (top-p sampling instead of beam search).

# 2. Method agnostic to base model
We would also like to note that our proposed approach is agnostic to the base model, and any neural synthesizer that has trainable parameters and outputs a sample of programs consistent with the the given examples can be used as the base model. The only other requirement is a speaker model that can sample examples consistent with a program (and prior examples, if applicable). If we use a different base synthesizer, we would have to compare that synthesizer with the version of the it trained on pragmatic examples chosen by our method.

We chose the RobustFill-style base synthesizer for its simplicity and flexibility – it makes minimal assumptions about the structure of the examples and the DSL – and showed that even an approach that makes minimal assumptions can benefit from our method. However, our method is compatible with other approaches such as DeepCoder (Balog et al., 2017), REPL (Ellis et al., 2019), DreamCoder (Ellis et al., 2021), etc. if we have a speaker model to generate examples.

# 3. Symbolic regular expression synthesizer
We also ran an experiment in the replay setting using the Regex+ model (Pertseva et al.) as the synthesizer. This is a domain-specific, fully symbolic version space algebra-based approach that incorporates a form of pragmatic reasoning similar to our speaker model. Even though the Regex+ synthesizer has a Consistent-10 metric^ of over 90% indicating that it is adept at finding regexes consistent with the examples, it achieves a Top-1 score of 0. The criteria we use to evaluate a pragmatic synthesizer is its ability to recover the target program exactly, since this suggests that the model is able to disambiguate user intent successfully. Finding a program that is consistent with the given examples is insufficient as many programs other than what the user intends may be consistent with a few given examples.

The low Top-1 score of Regex+ could be due to a couple of reasons:
- The Regex+ synthesizer accepts only positive examples
- Since we used the DSL out of the box, it differs from our DSL in not directly having access to some constructs that our model does such as `R{n, m}` which specifies between _n_ and _m_ occurrences of the unit `R`

^ Consistent-10 checks satisfying the examples analogous to Top-10

## References
- Jacob Devlin, et al. RobustFill: Neural program learning under noisy I/O. In ICML 2017.
- Matej Balog, et al. Deepcoder: Learning to write programs. In ICLR, 2017.
- Kevin Ellis, et al. 2019. Write, execute, assess: program synthesis with a REPL. NeurIPS
- Kevin Ellis, et al. 2021. DreamCoder: bootstrapping inductive program synthesis with wake-sleep library learning. In PLDI 2021.
- Elizaveta Pertseva, et al. Regex+: Synthesizing regular expressions from positive examples. 11TH Workshop on Synthesis.

---

### Author Response · Authors · 2023-11-20
**Preliminary DreamCoder results**

We also attempted to run our experiment with DreamCoder, which serves the same role as the literal synthesizer in our experiments. We use the same set up as the regular expression experiment in Ellis et al. (2021) (which supports only positive examples). Given the time constraints of the author response period we only have results from the first iteration of training. We note that DreamCoder gets a Consistent-10^ performance of 50%, it also achieves a Top-1 rating of 0. We expect the consistency metric to increase over iterations of training, but expect issues to persist with recovering the target program. This could again be (as with Regex+) that DreamCoder does not have direct access to some constructs such as `R{n, m}` which specifies between _n_ and _m_ occurrences of the unit `R`.

For example, given the examples [(SFGBHq, ✓), (AGq, ✓)] for the target [A-Z]{2,}(q|z), DreamCoder's guesses include `[A-Z]*q` and `[A-Z]*[a-z]`. Since DreamCoder does not support numeric ranges, expressing constructs such as at "least 2 of" leads to higher description length programs which are not enumerated first. Additionally, the lack of support for negative examples means that an example satisfying `[A-Z](q|z)` but not `[A-Z]{2,}(q|z)` (such as Aq) cannot be explicitly provided. However, we note that positive examples have still been found to be an effective means of describing regular expressions in prior work (Vaithilingam et al, 2023). The low scores thus point to the fact that most program synthesizers do not reason about the ambiguity of specifications, resulting in inability to recover the target program from among a number of consistent programs.

^ Consistent-10 metric is the analog of Top-10 which checks not whether one of the top 10 programs has the same semantics as the target, but instead that it is consistent with the given example. The fraction of tasks where this happens is averaged over the course of 10 turns as with the Top-10 metric.

### References
- Priyan Vaithilingam, Yewen Pu, and Elena L. Glassman. The usability of pragmatic communication in regular expression synthesis, 2023.
- Kevin Ellis, et al. 2021. DreamCoder: bootstrapping inductive program synthesis with wake-sleep library learning. In PLDI 2021.

---

### Meta-Review · Area_Chair_8DRb · 2023-12-01

**Metareview:**

This paper presents a method for program synthesis of regexes by example. It uses a listener and a speaker neural network model building on the Rational Speech Acts model, amortizing the cost of building these models and employing self-play. The presented method significantly improves over baselines, including LLMs.

### Strengths
- Clear and novel method on a practically useful problem.
- An interesting self-play method.
- Good improvements over baselines.

### Weaknesses
- It is unclear how this method would work on non-regex programs, with a vastly larger search spaces and input/output data types.

**Justification For Why Not Higher Score:**

While this work is interesting, its broader applicability to the community is unclear.

**Justification For Why Not Lower Score:**

This work is technically correct, interesting, and relevant. There is no reason to reject it.

---

### Decision · Program_Chairs · 2024-01-16

Accept (poster)